# California earthquake insurance unpopularity: the issue is the price, not the risk perception.

Adrien Pothon[1,2], Philippe Gueguen[1], Sylvain Buisine[2], and Pierre-Yves Bard[1]

[1]ISTerre, Université de Grenoble-Alpes/Université de Savoie Mont Blanc/CNRS/IRD/IFSTTAR, Grenoble, France
[2]AXA Group Risk Management, Paris, France

**Correspondence:** Adrien Pothon (adrien.pothon@univ-grenoble-alpes.fr)

**Abstract.** Despite California is a highly seismic prone region, most of homeowners are not covered against this risk. This study analyses the reasons for homeowners to purchase or not an insurance to cover earthquake losses, with application in California. A dedicated database is built from 18 different data sources about earthquake insurance, gathering data since 1921. A new model is developed to assess the take-up rate based on the homeowners' risk awareness and the average annual insurance premium amount. Results suggest that only two extreme situations would lead all owners to cover their home with insurance: (1) a widespread belief that a devastating earthquake is imminent, or alternatively, (2) a massive decrease in the average annual premium amount by a factor exceeding 6 (from $980 to $160, USD 2015). Considering the low likelihood of each situation, we conclude from this study that new insurance solutions are necessary to fill the protection gap.

## 1 Introduction

Since 2002, the rate of homeowners insured against earthquake risk in California has never exceeded 16%, according to data provided by the California Department of Insurance (CDI 2017b, California Insurance Market Share Reports). Such a low rate is surprising in a rich area, prone to earthquake risk. Consequently, several studies have already investigated homeowners' behaviour regarding earthquake insurance in California, in order to identify people who might have an interest in purchasing an earthquake insurance and to understand why they do not so. They have put in light that three main variables have been observed as decisive in purchasing an earthquake insurance: the premium amount, the socio-economic background and the risk perception (Kunreuther et al. 1978; Palm and Hodgson 1992a; Wachtendorf and Sheng 2002). For the premium amount, both a survey conducted by Meltsner (1978) and the statistics collected from insurance data by Buffinton (1961) and Latourrette (2010) show, as expected, a negative correlation between the premium amount and the insurance adoption. Nevertheless, even uninsured homeowners tend to overestimate the loss they would face in case of a major event and feel vulnerable regardless of their income level (Kunreuther et al. 1978; Palm and Hodgson 1992a). As they are not expecting much from federal aids, they do not know how they will recover (Kunreuther et al. 1978).

Consequently the decision to purchase an insurance to cover earthquake losses is uncorrelated to the income level (Kunreuther et al. 1978; Wachtendorf and Sheng 2002). Several other socio-economic factors (e.g. the duration of residence, the neighbours' behaviour, the communication strategies of mass media, real estate agencies and insurance companies) have an impact on the

risk perception (Meltsner 1978; Palm and Hodgson 1992a; Lin 2013).

Pricing methods used in the real estate market, do not take into account seismic risk. Indeed, based on data from the World Housing Encyclopedia, Pothon et al. (2018) have shown that building construction price is not correlated to the seismic vulnerability rating. Moreover, earthquake insurance is not mandatory in California for residential mortgage (CDI 2019, Information Guides, Earthquake Insurance). Still, lenders for commercial mortgages, are used to requiring an earthquake insurance only when the probable maximum loss is high (Porter et al. 2004). However, Porter et al. (2004) have shown that taking into account earthquake risk for calculating the building's net asset value can have a significant impact in earthquake prone areas like California.

Also, insured people against earthquake risk can receive a compensation lower than the loss incurred because most of earthquake insurance policies in California include a deductibles amount (i.e. at the charge of the policyholder) and are calibrated based on a total reconstruction cost, declared by the policyholder. As reported by Marquis et al. (2017) after the 2010–2011 Canterbury (NZL) earthquakes, this amount can be inadequate for the actual repairing costs. Both high deductibles amount (Meltsner 1978; Palm and Hodgson 1992b; Burnett and Burnett 2009) and underestimated total reconstruction cost (Garatt and Marshall 2003) can make insured people felt unprotected after an earthquake, disreputing this kind of insurance.

Since homeowners are aware of the destructive potential of a major earthquake, the risk perception reflects their personal estimate of the occurrence (Kunreuther et al. 1978; Wachtendorf and Sheng 2002). Indeed, most of them disregard this risk because it is seldom, even if destructive earthquake experiences foster insurance underwriting during the following year (Buffinton 1961; Kunreuther et al. 1978; Meltsner 1978; Lin 2015). Consequently, earthquake insurance purchasing behaviour is correlated to personal understanding of seismic risk and is not only related to scientific-based hazard level (Palm and Hodgson 1992a; Palm 1995; Wachtendorf and Sheng 2002).

Limits in earthquake insurance consumption being identified, the next step is to assess the contribution of each factor to the take-up rate. Kunreuther et al. (1978) first proposed a sequential model showing that the individual's purchasing behaviour depends on personal risk perception, premium amount and knowledge of insurance solutions. In the same study (Kunreuther et al. 1978), a numerical model has been also developed, at the same granularity, failing however to reproduce accurately the observed behaviours. According to the authors, the model failed because many surveyed people had a lack of knowledge in existing insurance solutions or were unable to quantify the risk. Models published later (Latourrette et al. 2010; Lin 2013; Lin 2015) assessed the take-up rate by zipcode and used demographic variables to capture the disparity in insurance solutions knowledge. Nevertheless because of lack of data, they assumed the premium amount or the risk perception as constant.

The main objective of this study is to introduce a new take-up rate model for homeowners at the scale of California State. Such spatial resolution allows working on most of data available in financial reports, which is numerous enough to use both the premium amount and the risk perception as variables. Contrasting from previous studies, focusing on risk pricing (Yucemen 2005; Petseti and Nektarios 2012; Asprone et al. 2013), this study takes into account the homeowners' risk perception and behaviour. This shift in point of view approach changes the main issue from: what should be the price of earthquake insurance considering the risk level? to: What is the acceptable price for consumers to purchase an earthquake insurance cover? Despite results are at the state level, they bring a new framework to model earthquake insurance consumption which allow to quantify

the gap between premium amount and homeowners' willingness to pay, depending on the homeowners' risk awareness. Last, this study is also innovative by modelling separately the contribution of the risk perception and the premium amount to the level of earthquake insurance consumption.

In the first section, data is collected from several sources, processed and summarized in a new database. A take-up rate model is then developed by introducing the two following explanatory variables: the average premium amount and the subjective annual occurrence probability, defined as the risk perceived by homeowners of being affected by a destructive earthquake in one year. The perceived annual probability of occurrence is then studied since 1926, in the next section. Finally, the last section is dedicated to analysing the current low earthquake insurance take-up rate for California homeowners.

## 2 Data collection and processing

Developing such a model, even at the State level, faces a first challenge in data collection. Despite California earthquake insurance market has been widely analysed, data volume before the 1990s remains low and extracted from scientific and mass media publications which refer to original data sources that are no longer available. As a consequence, data description is often sparse, incomplete and values can be subject to errors or bias. Consequently, data quality of each data set has to be assessed. Here, it is done on the basis of the support (scientific publication or mass media), the data description (quality of information on the data type and the collecting process) and the number of records. The datasets with the highest quality are used and presented in Table 1 (values are available in the electronic supplement).

This study focuses on the following variables about earthquake insurance policies:

- the total written premium ($W_N$), corresponding to the total amount paid by policyholders to insurance companies during the year $N$;

- the take-up rate ($t_N$), defined as the ratio between the number of policies with an earthquake coverage ($Nb_N$) and those with a fire coverage ($Fi_N$);

- the annual average premium ($P_N$), equal to $W_N$ over $Nb_N$;

- the average premium rate ($p_N$) equal to the ratio between the annual average premium amount $P_N$ and the average value of the good insured (here a house), later referred as the average sum insured ($ASI_N$).

All these variables can be calculated for any set of earthquake insurance policies. Insurance companies usually classify their products as follows: the Residential line of business ($Res$) corresponds to insurance policies covering personal goods (e.g. the house, condominium, mobile home, jewellery, furniture). The Homeowners line of business corresponds to the insurance policies included in the Residential line of business but dedicated to homeowners. As an earthquake insurance cover can be either a guarantee included in a wider policy (e.g. covering also fire or theft risks) or issued in a stand-alone policy, the Earthquake line of business ($Eq$) classifies all insurance policies covering exclusively the earthquake risk. Some of insurance policies within the Earthquake line of business are dedicated to professional clients, and do not belong to the Residential line of business. For

**Table 1.** Raw data collected about earthquake insurance. Labels in italic are not extracted from publications but have been inferred by cross-checking with other sources. The data quality scale is: A (good): Scientific publication with methodology explained; B (acceptable): Scientific publication without details on the methodology; C (weak): mass media. CPI: Consumer Price Index; Eq: Earthquake; Ho: Homeowners; LA: Los Angeles; LOB: Line of Business; Res: Residential; SF: San Francisco.

| Variable | Metric | LOB | Period | Data quality | Source |
|---|---|---|---|---|---|
| Average premium ($P_N$) | Amount | Ho; Res | 1996-2016 | A | CDI |
| | Rate | Ho | 1972 | A | Kunreuther et al. (1978) |
| | Rate | Ho | 1991 | C | Shiver Jr (1991) |
| | Rate | *Res* | 1926 - 1930 | A | Freeman (1932) |
| | Rate | Res | 1956 | B | Buffinton (1961) |
| | Amount | Res | 1976 | A | Steinbrugge et al. (1980) |
| | Amount | Res | 1977 | A | Steinbrugge et al. (1980) |
| | Amount | Res | 1978 | A | Steinbrugge et al. (1980) |
| | Amount | Res | 1992 | A | Lagorio et al. (1992) |
| | Annual Variation | Res | 1995 | C | Mulligan (1994) |
| Take-up rate ($t_N$) | Value | Ho | 1972 | A | Kunreuther et al. (1978) |
| | Value | Ho | 1989 | C | Kunreuther (2015) |
| | Value | Ho | 1990 | A | Garamendi et al. (1992) |
| | Value | Ho | 1992 | A | Lagorio et al. (1992) |
| | Value | Ho | 1993 | C | Kunreuther (2015) |
| | Value | Ho | 1994 | A | Roth (1997) |
| | Value | Ho; Res | 1996 - 2016 | A | CDI |
| | Value | Res | 1926 - 1930 | A | Freeman (1932) |
| | Value | Res | 1956 | B | Buffinton (1961) |
| | Value | Res | 1971 | B | Roth (1997) |
| | Value | Res | 1976 | B | Kunreuther et al. (1992) |
| | Value | Res | 1978 | A | Steinbrugge et al. (1980) |
| | Annual Variation | Res | 1991 | B | Kunreuther et al. (1992) |
| | Value | Res | 1995 | B | Jones et al. (2012) |
| Total written premium ($W_N$) | *Written amount* | Ho; Res | 1996 - 2016 | A | CDI |
| | Written amount | Eq | 1992 - 2016 | A | CDI |
| Total earned premium ($E_N$) | Earned amount | *Eq* | 1921 - 1929 | A | Freeman (1932) |
| | *Earned amount* | Eq | 1930 - 1969 | B | Meltsner (1978) |
| | Earned amount | *Eq* | 1970 - 1991 | A | Jones et al. (2012) |
| | Earned amount | Eq | 1992 - 2016 | A | CDI |
| Number of earthquake policies ($Nb_N$) | Number | Ho | 1990 | A | Garamendi et al. (1992) |
| | Number | Ho | 1994 | A | Roth (1997) |
| Average sum insured ($ASI_N$) | Amount | Ho | 2006 - 2016 | A | CDI |
| Number of fire policies ($Fi_N$) | Amount | Res | 1996 - 2016 | A | CDI |
| Socio-economic indicators ($CPI_N$; $Pop_N$; $RBCI_N$) | CPI Urban LA + SF | - | 1921 - 2016 | A | U.S. Department of Labor |
| | Population census | - | 1921 - 2016 | A | U.S. Census Bureau |
| | Real Building Cost Index | - | 1921 - 2015 | A | Shiller (2015) |

clarity, variables hold the acronym of the line of business when they are not related to the Homeowners line of business.

Data for all the variables listed is available only since 1996 (Table 1). In order to expand the historical database, they are also estimated in an indirect way for the following periods: 1926-1930, 1956, 1971-1972, 1976-1978 and 1989-1995, leading to consider additional data (Table 1) and to use linear regressions (Figure 1).

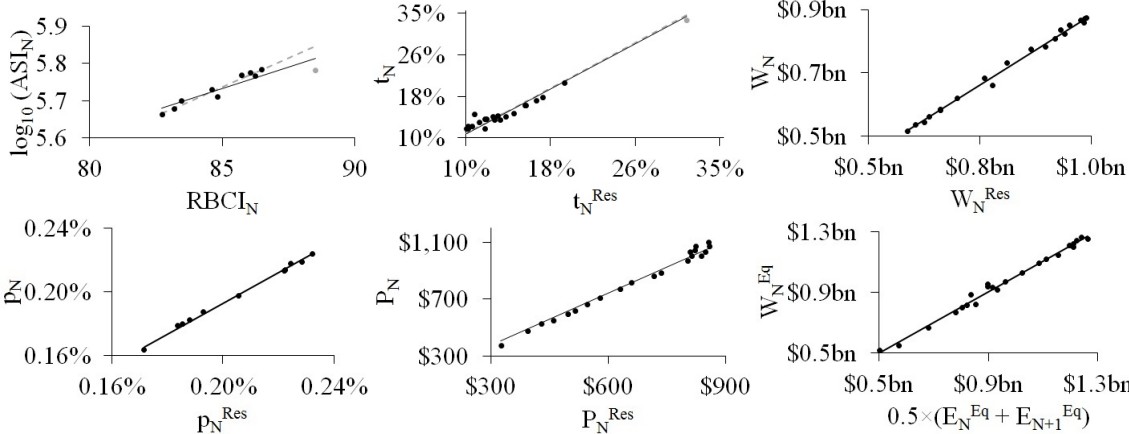

**Figure 1.** Fits of the linear regressions for: a) the average sum insured and the Real Building Cost Index, b) the average premium rate between the Residential and the Homeowners lines of business, c) the take-up rate between the Residential and the Homeowners lines of business, d) the average premium amount between the Residential and the Homeowners lines of business, e) the total written premium amount between the Residential and the Homeowners lines of business and f) the written and the earned premium amounts. Grey points on a) and c) represent the extreme values, affecting the linear regression from the solid black line to the dashed grey line, when removed. Financial values are in USD 2015.

By definition of the average premium rate ($p_N$), $P_N$ can be approximated by the product of $p_N$ and the average sum insured ($ASI_N$). The latter is estimated from the Real Building Cost Index ($RBCI_N$), which is an economic index (base: 31/12/1979=100) capturing the evolution of the cost of building construction works (Shiller 2008). The following linear regression has been calibrated on data from Table 1 between 2006 and 2015:

$$\log_{10}(ASI_N) = 0.0233 \times RBCI_N + 3.7554 \qquad \left(R^2 = 0.84 \quad \text{Figure 1a}\right) \tag{1}$$

where $ASI_N$ is in USD 2015 and $R^2$ is the coefficient of determination. When the average premium rate corresponds to the Residential line of business ($p_N^{Res}$), it can be converted into $p_N$ with the following linear regression built on the CDI database between 2006 and 2016 (Table 1):

$$p_N = 0.96 \times p_N^{Res} \qquad \left(R^2 = 1 \quad \text{Figure 1b}\right) \tag{2}$$

Using again the CDI database between 1996 and 2016 (Table 1), the following linear regressions have also been developed between $P_N$, $t_N$ and $W_N$ and the corresponding metrics for the Residential line of business $\left(P_N^{Res}, t_N^{Res} \text{ and } W_N^{Res}\right)$:

$$t_N = 1.08 \times t_N^{Res} \qquad \left(R^2 = 0.95 \quad \text{Figure 1c}\right) \tag{3}$$

$$P_N = 1.24 \times P_N^{Res} \qquad \left(R^2 = 0.99 \quad \text{Figure 1d}\right) \tag{4}$$

$$W_N = 0.88 \times W_N^{Res} \qquad \left( R^2 = 0.99 \quad \text{Figure 1e} \right) \tag{5}$$

Before 1996, the total premium amount for earthquake insurance, mentioned in CDI reports, was about the Earthquake line of business, and in terms of total earned premium ($E_N$). While the total written premium ($W_N$) corresponds to the total amount of premium paid by policyholders to insurance companies during the year $N$, $E_N$ is the amount of premium used to cover the risk during the year $N$. To illustrate these two definitions, Figure 2 takes the example of an insurance policy for which the annual premium is paid every March, $1^{\text{st}}$. As the amount received at year $N$ by the insurance company ($W_N$) covers the risk until March $1^{\text{st}}$, $N+1$, the insurance company can use only 75% of $W_N$ during the year $N$ (9 months over 12). By adding the 25% of $W_{N-1}$, this makes the total earned premium $E_N$, for the year $N$. To estimate $W_N^{EQ}$ from $E_N^{Eq}$ for the Earthquake line

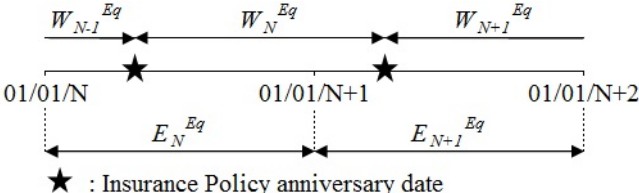

**Figure 2.** Illustration of the difference between the earned and the written premium amount.

of business, the following relationship has been defined based on data from the CDI database and over the period 1992-2016 (Table 1):

$$W_N^{Eq} = \frac{E_N^{Eq} + E_{N+1}^{Eq}}{2} \qquad \left( R^2 = 0.99 \quad \text{Figure 1f} \right) \tag{6}$$

About the difference between $W_N^{Eq}$ and $W_N^{Res}$, despite data (Table 1) shows a significant difference $\left( W_N^{Eq} - W_N^{Res} = \$250\text{m.} \right)$ since 1996, this study assumes that $W_N^{Eq}$ is equal to $W_N^{Res}$ until 1995, i.e.:

$$W_N^{Res} \approx \begin{cases} W_N^{Eq} & \text{if} \quad N \le 1995 \\ W_N^{Eq} - \$249,188,671 & \text{else} \end{cases} \tag{7}$$

This strong assumption was used by Garamendi et al. (1992) and was verified for the loss after the 1994 Northridge earthquake reported by Eguchi et al. (1998).

Last, when variables cannot be inferred from other variables, they are estimated based on the annual variation for $P_N^{Res}$:

$$P_N^{Res} = P_{N-1}^{Res} \times \Delta_{P_{N-1}^{Res}}^{P_N^{Res}} \tag{8}$$

and the biennial variation for $t_N^{Res}$:

$$t_N^{Res} = t_{N-2}^{Res} \times \Delta_{t_{N-2}^{Res}}^{t_N^{Res}} \tag{9}$$

where $\Delta_X^Y$ is equal to the ratio $Y$ over $X$. Furthermore, the variation of the total written premium for the Residential line of business $\left(\Delta_{W_{N-1}^{Res}}^{W_N^{Res}}\right)$ is linked to $\Delta_{P_{N-1}^{Res}}^{P_N^{Res}}$, $\Delta_{t_{N-1}^{Res}}^{t_N^{Res}}$ and $\Delta_{Fi_{N-1}^{Res}}^{Fi_N^{Res}}$ as follows:

$$
\begin{aligned}
\Delta_{W_{N-1}^{Res}}^{W_N^{Res}} &= \frac{W_N^{Res}}{W_{N-1}^{Res}} \\
&= \frac{P_N^{Res} \times Nb_N^{Res}}{P_{N-1}^{Res} \times Nb_{N-1}^{Res}} \\
&= \frac{P_N^{Res} \times Fi_N^{Res} \times t_N^{Res}}{P_{N-1}^{Res} \times Fi_{N-1}^{Res} \times t_{N-1}^{Res}} \\
&= \Delta_{P_{N-1}^{Res}}^{P_N^{Res}} \times \Delta_{Fi_{N-1}^{Res}}^{Fi_N^{Res}} \times \Delta_{t_{N-1}^{Res}}^{t_N^{Res}}
\end{aligned}
\tag{10}
$$

As we have no data regarding the annual variation of the number of fire insurance policy for the residential line of business $\left(\Delta_{Fi_{N-1}^{Res}}^{Fi_N^{Res}}\right)$ before 1996 (Table 1), it is assumed as equal to the variation of the California Population $Pop_N$:

$$
\Delta_{Fi_{N-1}^{Res}}^{Fi_N^{Res}} \approx \Delta_{Pop_{N-1}}^{Pop_N}
\tag{11}
$$

Figures 1 and 3 illustrate the goodness-of-fit of the regressions developed (Equations 1 to 6) and the variations of the population compared to the number of Fire insurance policies between 1996 and 2016 (Equation 11), respectively.

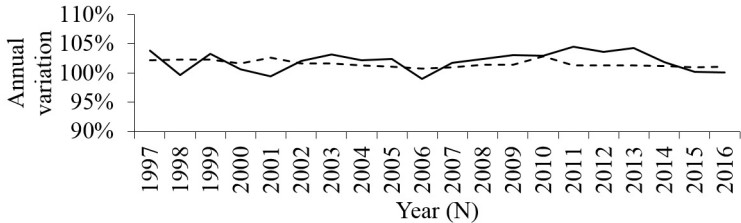

**Figure 3.** Comparison between the annual variations of the number of Fire insurance policies for the Residential line of business (solid line) and the California population (dashed line).

Finally, values of $P_N$, $t_N$ and $W_N$ collected (Table 1) and estimated in this study (Equations 1 to 11) are aggregated into a new database presented in Table 2. Financial amounts are converted into USD 2015 using the Consumer Price Index (U.S. Dept. of Labor 2017). The data quality of each variable is also assessed at the weakest data quality used (Table 1), downgraded of as much levels as the number of equations used to assess it (listed in the column Equation Id). For example, the annual average premium $P_{1926}$ has been calculated from the premium rate for the Residential line of business $p_{1926}^{Res}$ (Table 1) and the Equations 1 and 4. The associated Data Quality is C (Table 2) because the Data Quality of $p_{1926}^{Res}$ is A (Table 1), downgraded of two levels for the two equations used.

The model developed in the next section uses this new database to estimate the take-up rate for the California homeowners ($t_N$) from the average premium amount ($P_N$) and another variable capturing the relative earthquake risk awareness.

**Table 2.** The California Homeowners Earthquake Insurance database developed for this study. Data quality scale is: A (good): all data from scientific publications with methodology explained; (B) acceptable: at least one data from a scientific publication without details on the methodology or one processing has been applied; (C) weak: data is uncertain. Financial amounts are in USD 2015. The total written premium is given in million USD. Column names are Q: Data Quality; E: Equation Id. Abbreviations are: $E_1 = \{3; 6; 7; 10\}$; $E_2 = \{5; 6; 7\}$.

| $N$ | $t_N$ | Q | E | $P_N$ | Q | E | $W_N$ | Q | E | $N$ | $W_N$ | Q | E | $N$ | $W_N$ | Q | E |
|---|---|---|---|---|---|---|---|---|---|---|---|---|---|---|---|---|---|
| 1926 | 5% | B | 3 | 612 | C | 1;4 | 3 | C | $E_2$ | 1931 | 1 | C | $E_2$ | 1973 | 12 | C | $E_2$ |
| 1927 | 4% | B | 3 | 809 | C | 1;4 | 2 | C | $E_2$ | 1932 | 1 | C | $E_2$ | 1974 | 13 | C | $E_2$ |
| 1928 | 6% | B | 3 | 441 | C | 1;4 | 2 | C | $E_2$ | 1933 | 1 | C | $E_2$ | 1975 | 15 | C | $E_2$ |
| 1929 | 7% | B | 3 | 339 | C | 1;4 | 2 | C | $E_2$ | 1934 | 1 | C | $E_2$ | 1979 | 33 | C | $E_2$ |
| 1930 | 8% | B | 3 | 287 | C | 1;4 | 2 | C | $E_2$ | 1935 | 1 | C | $E_2$ | 1980 | 44 | C | $E_2$ |
| 1956 | 3% | C | 3 | 1101 | C | 1;4 | 5 | C | $E_2$ | 1936 | 1 | C | $E_2$ | 1981 | 54 | C | $E_2$ |
| 1971 | 8% | C | 3 | 285 | C | 1;4;$E_1$ | 7 | C | $E_2$ | 1937 | 1 | C | $E_2$ | 1982 | 65 | C | $E_2$ |
| 1972 | 1% | A | - | 2228 | B | 1 | 10 | C | $E_2$ | 1938 | 1 | C | $E_2$ | 1983 | 76 | C | $E_2$ |
| 1976 | 3% | C | 3 | 670 | B | 4 | 18 | C | $E_2$ | 1939 | 1 | C | $E_2$ | 1984 | 108 | C | $E_2$ |
| 1977 | 6% | C | $E_1$ | 624 | B | 4 | 21 | C | $E_2$ | 1940 | 1 | C | $E_2$ | 1985 | 158 | C | $E_2$ |
| 1978 | 8% | B | 3 | 576 | B | 4 | 26 | C | $E_2$ | 1941 | 1 | C | $E_2$ | 1986 | 195 | C | $E_2$ |
| 1989 | 22% | C | - | 586 | C | 4;10;$E_2$ | 360 | C | $E_2$ | 1942 | 1 | C | $E_2$ | 1987 | 244 | C | $E_2$ |
| 1990 | 25% | A | - | 551 | C | $E_2$ | 405 | C | $E_2$ | 1943 | 2 | C | $E_2$ | 1988 | 307 | C | $E_2$ |
| 1991 | 24% | C | 3;9 | 506 | C | 1 | 480 | C | $E_2$ | 1944 | 2 | C | $E_2$ | | | | |
| 1992 | 100% | A | - | 77 | B | 4 | 519 | C | 5;7 | 1945 | 2 | C | $E_2$ | | | | |
| 1993 | 37% | C | - | 348 | C | 4;5;$E_1$ | 550 | C | 5;7 | 1946 | 3 | C | $E_2$ | | | | |
| 1994 | 31% | A | - | 511 | C | 5;7 | 668 | C | 5;7 | 1947 | 3 | C | $E_2$ | | | | |
| 1995 | 22% | C | 3 | 842 | C | 4;8;$E_2$ | 883 | C | 5;7 | 1948 | 3 | C | $E_2$ | | | | |
| 1996 | 33% | A | - | 606 | A | - | 778 | A | - | 1949 | 3 | C | $E_2$ | | | | |
| 1997 | 21% | A | - | 753 | A | - | 606 | A | - | 1950 | 3 | C | $E_2$ | | | | |
| 1998 | 18% | A | - | 810 | A | - | 589 | A | - | 1951 | 4 | C | $E_2$ | | | | |
| 1999 | 18% | A | - | 821 | A | - | 626 | A | - | 1952 | 4 | C | $E_2$ | | | | |
| 2000 | 17% | A | - | 850 | A | - | 637 | A | - | 1953 | 4 | C | $E_2$ | | | | |
| 2001 | 16% | A | - | 851 | A | - | 661 | A | - | 1954 | 4 | C | $E_2$ | | | | |
| 2002 | 15% | A | - | 888 | A | - | 661 | A | - | 1955 | 4 | C | $E_2$ | | | | |
| 2003 | 14% | A | - | 931 | A | - | 661 | A | - | 1957 | 5 | C | $E_2$ | | | | |
| 2004 | 14% | A | - | 990 | A | - | 761 | A | - | 1958 | 5 | C | $E_2$ | | | | |
| 2005 | 12% | A | - | 1020 | A | - | 699 | A | - | 1959 | 5 | C | $E_2$ | | | | |
| 2006 | 14% | A | - | 1035 | A | - | 810 | A | - | 1960 | 6 | C | $E_2$ | | | | |
| 2007 | 14% | A | - | 1025 | A | - | 865 | A | - | 1961 | 5 | C | $E_2$ | | | | |
| 2008 | 14% | A | - | 1092 | A | - | 931 | A | - | 1962 | 6 | C | $E_2$ | | | | |
| 2009 | 14% | A | - | 1128 | A | - | 952 | A | - | 1963 | 6 | C | $E_2$ | | | | |
| 2010 | 14% | A | - | 1148 | A | - | 978 | A | - | 1964 | 5 | C | $E_2$ | | | | |
| 2011 | 13% | A | - | 1157 | A | - | 989 | A | - | 1965 | 5 | C | $E_2$ | | | | |
| 2012 | 12% | A | - | 1101 | A | - | 920 | A | - | 1966 | 5 | C | $E_2$ | | | | |
| 2013 | 12% | A | - | 1073 | A | - | 897 | A | - | 1967 | 5 | C | $E_2$ | | | | |
| 2014 | 12% | A | - | 1092 | A | - | 940 | A | - | 1968 | 6 | C | $E_2$ | | | | |
| 2015 | 12% | A | - | 1103 | A | - | 984 | A | - | 1969 | 6 | C | $E_2$ | | | | |
| 2016 | 15% | A | - | 980 | A | - | 986 | A | - | 1970 | 5 | C | $E_2$ | | | | |

## 3  Model development for the period 1997-2016

In order not to presume a linear trend between the consumers' behaviour and explanatory variables (e.g. the average premium amount), this study refers to the Expected Utility theory (Von Neumann and Morgenstern 1944) instead of statistical linear models. The Expected Utility theory is a classical framework in Economic science to model the economical choices of a consumer, depending on several variables like the wealth or the risk aversion (Appendix A). Here the application field is the California homeowners instead of a single consumer. Within this framework, homeowners are assumed rational and taking decisions in order to maximize their utility. Furthermore, their relative risk aversion is considered constant whatever the wealth because the decision to purchase or not an earthquake insurance is independent from the income level (Kunreuther et al. 1978; Wachtendorf and Sheng 2002). One of the most used utility functions ($U$) is (Holt and Laury 2002):

$$U\big(g_N(t_N)\big) = \frac{g_N(t_N)^{1-\beta}}{1-\beta} \tag{12}$$

where $g_N$ and $\beta$ are the wealth function at year $N$ and the risk profile controlling the risk aversion level (the larger $\beta$, the higher the aversion), respectively.

Then, the average household's capital $K$ is assumed in this study equal to the average sum insured ($ASI_{2015}$) since the earthquake insurance consumption is uncorrelated to the wealth (Kunreuther et al. 1978; Wachtendorf and Sheng 2002) and equal to \$604,124 (USD 2015). Consequently, the wealth of uninsured homeowners is $g_N = K$, if no damaging earthquake occurs. Regarding the loss estimation, homeowners are mostly concerned by destructive earthquakes, defined as earthquakes which can potentially damage their home, and tend to overestimate the impact (Buffinton 1961; Kunreuther et al. 1978; Meltsner 1978; Lin 2015). Furthermore, according to Kunreuther et al. (1978), insured people believe that they are fully covered in case of loss. Therefore, we assume that only uninsured homeowners expect to incur a loss equal to $K$ after a damaging earthquake. This lead us to model that uninsured homeowners expect a wealth at $g_N = 0$ after a damaging earthquake. About insured homeowners, we assume that they expect to have a constant wealth at $g_N = K - P_N$, since they believe to be fully covered in case of losses induced by an earthquake. Table 3 summarizes these three wealth levels, based on the occurrence of a devastating earthquake and the insurance cover. Finally, taking into account the share of insured homeowners ($t_N$) and the results in Table 3, $g_N$ can

**Table 3.** The four different wealth levels of a homeowner, considered in this study, according on the occurrence of a damaging earthquake and the insurance cover. The quantities $t_N$ and $1 - t_N$ represent the share of homeowners insured and not insured, respectively.

| Occurrence of a damaging earthquake | Insured ($t_N$) | Not insured ($1 - t_N$) |
|---|---|---|
| Yes | $K - P_N$ | 0 |
| No | $K - P_N$ | $K$ |

written for all homeowners:

$$g_N(t_N) = K - t_N \times P_N - K \times (1 - t_N) \times \mathcal{B}_{EQ}(r_N) \tag{13}$$

where $\mathcal{B}_{EQ}(r_N)$ is a random variable following the Bernoulli distribution (i.e. equal to 1 if a damaging earthquake occurs during the year $N$ and 0 otherwise). The parameter $r_N$, called the subjective annual occurrence probability, controls the homeowners' risk perception through the perceived probability of being affected by a destructive earthquake during the year $N$ (Kunreuther et al. 1978; Wachtendorf and Sheng 2002). As homeowners want to maximize their utility, $t_N$ is solution of:

$$t_N^{Estimated} = \underset{0 \leq t_N \leq 1}{\mathrm{argmax}} \ \mathbb{E}\left[U(g_N(t_N))\right] \tag{14}$$

where argmax stands for the argument of the maxima function (i.e. which returns the value of $t_N$ which maximizes the quantity $\mathbb{E}\left[U(g_N(t_N))\right]$) and $\mathbb{E}$ is the expected value of $U(g_N(t_N))$ which depends from the random variable $\mathcal{B}_{EQ}(r_N)$ (Equation 13). Assuming that homeowners are risk averse (i.e. $\beta > 0$) and noticing that $\frac{P_N}{K}$ is very small compared to 1, $t_N^{Estimated}$ is shown (Appendix B) to be equal to:

$$t_N^{Estimated} = \min\left[\left(\frac{r_N}{1-r_N} \times \frac{K}{P_N}\right)^{\frac{1}{\beta}}; 100\%\right] \tag{15}$$

As a first step, the model is calibrated with data on the period 1997-2016 corresponding to the whole activity period of the California Earthquake Authority (CEA) providing high quality data (Table 2). Created on December 1996, this is a non-profit, state managed, organization, selling residential earthquake policies (Marshall 2017). The stability of the insurance activity and the lack of devastating earthquake lead us to model parameters $\beta$ and $r_N$, capturing the homeowners' risk perception, by constant variables. They are estimated at $r_{1997;...;2016} = 0.027\%$ and $\beta = 0.93$, using the Least Squared method with the Generalized Reduced Gradient algorithm. The model (Equation 15) fits the observed data with a $R^2 = 0.79$ (Figure 4).

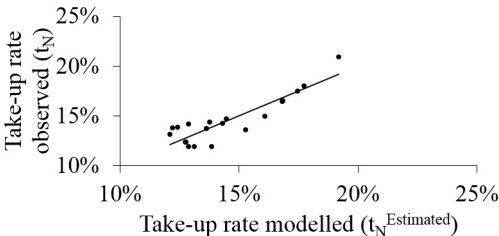

**Figure 4.** Fit between the take-up rates modelled and observed for the period 1997-2016.

Furthermore, the values of the two parameters are meaningful: homeowners are modelled "very risk averse" ($\beta = 0.93$), which is consistent with a high severity risk like an earthquake (Holt and Laury 2002). $r_N$ is also somehow consistent with the value that can be calculated from a hazard analysis, as presented later.

The model calibration between 1997 and 2016 is not appropriate for other periods corresponding to different seismic activity and insurance economic context. The next section investigates how the homeowners' risk perception has changed since 1926 in order to adapt the model to any period.

## 4  Evolution of the homeowners' risk perception since 1926

The homeowners' risk perception is controlled by both the risk materiality (what kind of earthquakes are expected to occur?) and the risk tolerance (how much homeowners are ready to loose?). The associated parameters in the model are $r_N$ and $\beta$, respectively. As we cannot differentiate the one from the other, $\beta$ is assumed constant in this study and all the variations of the homeowner's risk perception are passed on to $r_N$.

According to the model and the relationships developed (Equations 3, 4, 5, 10, 11 and 15), the variations of the total written premium amount per capita ($W_N/Pop_N$) depend on the variation of the average premium amount ($P_N$) and the subjective annual occurrence probability ($r_N$):

$$
\begin{aligned}
\frac{W_N/Pop_N}{W_{N-1}/Pop_{N-1}} &= \frac{\Delta^{W_N^{Res}}_{W_{N-1}}}{\Delta^{Pop_N}_{Pop_{N-1}}} \\
&= \Delta^{P_N}_{P_{N-1}} \times \Delta^{t_N}_{t_{N-1}} \\
&= \Delta^{P_N}_{P_{N-1}} \times \frac{\min\left[\left(\dfrac{r_N}{1-r_N} \times \dfrac{K}{P_N}\right)^{\frac{1}{\beta}}; 100\%\right]}{\min\left[\left(\dfrac{r_{N-1}}{1-r_{N-1}} \times \dfrac{K}{P_{N-1}}\right)^{\frac{1}{\beta}}; 100\%\right]}
\end{aligned}
\tag{16}
$$

Under the assumption that $t_N^{Estimated}$ has never reached 100% in the past and with $\beta = 0.93$, Equation 16 can be simplified as follows:

$$
\begin{aligned}
\frac{W_N/Pop_N}{W_{N-1}/Pop_{N-1}} &= \Delta^{P_N}_{P_{N-1}} \times \frac{\left(\dfrac{r_N}{1-r_N} \times \dfrac{K}{P_N}\right)^{\frac{1}{\beta}}}{\left(\dfrac{r_{N-1}}{1-r_{N-1}} \times \dfrac{K}{P_{N-1}}\right)^{\frac{1}{\beta}}} \\
&= \left(\frac{P_N}{P_{N-1}}\right)^{-0.07} \times \left(\frac{r_N \times (1-r_{N-1})}{r_{N-1} \times (1-r_N)}\right)^{1.07}
\end{aligned}
\tag{17}
$$

The contribution of $P_N/P_{N-1}$ is clearly marginal, leading to consider the regression, built on the variations of $W_N/Pop_N$ and $r_N$ between 1997 and 2016 (Table 2):

$$
\frac{r_N}{r_{N-1}} - 1 = 0.92 \times \left(\frac{W_N/Pop_N}{W_{N-1}/Pop_{N-1}} - 1\right) \qquad (\text{R}^2 = 1)
\tag{18}
$$

Equations 16 and 18 used and the goodness-of-fit are illustrated in Figures 5a and 5b, respectively. Considering that understanding the details of those variations is out of reach given the scarcity of data, this study focuses only on those above $15.5\%$ in absolute value, qualified hereafter as significant variations ($V_N$). Other variations are neglected and we assume they cancel each other out. Table 4 lists all of them, together with the most significant event occurred the same year, for the earthquake insurance industry. Despite these events (Table 4) could have led insurance companies to restrict or enlarge the number of earthquake insurance policies (Born and Klimaszewski-Blettner 2013), contributing in this way to $V_N$, this study focuses only

(a)

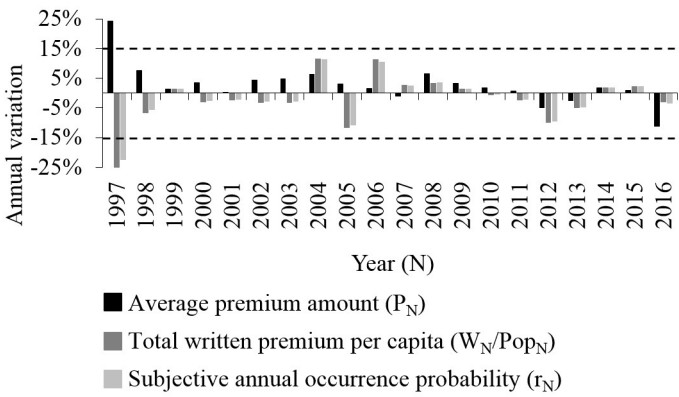

■ Average premium amount (P_N)
■ Total written premium per capita (W_N/Pop_N)
■ Subjective annual occurrence probability (r_N)

(b)

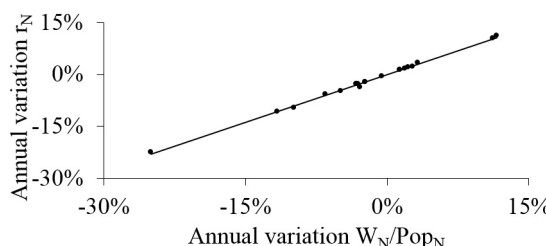

**Figure 5.** a) Variations of the average premium amount, the total written premium per capita and the subjective annual occurrence probability between 1997 and 2016. Variations are calculated compared to the previous year. Dashed lines represent the thresholds for a significant variation. b) Fit between the variation of the total written premium per capita and the subjective annual occurrence probability.

on the homeowners' risk perception variations. Then, the subjective annual occurrence probability ($r_N$) is estimated according to the following time series for $N \in [1926; 2016]$:

$$
\begin{cases}
r_{2016} & = 0.027\% \\
r_N & = \dfrac{r_{N+1}}{1 + V_{N+1}} \qquad \forall N \in [1926; 2015]
\end{cases}
\tag{19}
$$

The time series of $r_N$ is represented in Figure 6 and illustrates that some earthquakes (indicated by dots) have increased $r_N$ during the year as already published (Buffinton 1961; Kunreuther et al. 1978; Meltsner 1978; Lin 2015) but, more surprisingly, some others had no apparent impact, like the 1952 Kern County and the 1989 Loma Prieta earthquakes. The first one damaged over 400 unreinforced masonry buildings (Jones et al. 2012) but none of the buildings designed under the latest seismic codes (Geschwind 2001). Consequently, structural engineers claimed that new buildings were earthquake-proof and that no additional prevention measures were needed (Geschwind 2001). This message received a great echo among the population despite the

**Table 4.** Events expected to have significantly modified the homeowners' risk perception through the subjective annual occurrence probability. $V_N$ represent the variation compared to the previous year, with the sign '+' for an increase and '-' for a decrease.

| Category | Period | Major event occurred | Variation ($V_N$) |
|---|---|---|---|
| Earthquake | 1933 | Long Beach earthquake | +22% |
| | 1971-1972 | San Fernando earthquake | +65% |
| | 1979 | Imperial Valley & Coyote Lake earthquakes | +15% |
| | 1984-1988 | High seismic activity | +109% |
| | 1994-1995 | Northridge earthquake | +49% |
| Socio-economic | 1928 | Discredit of a major earthquake in Southern California | -20% |
| | 1931-1932 | Great Depression | -50% |
| | 1946 | Post World War II economic expansion | +22% |
| | 1984-1985 | Earthquake coverage mandatory offer law | +30% |
| | 1997 | Restricted mandatory earthquake coverage | -23% |

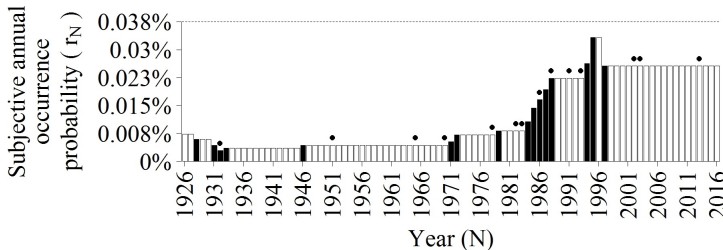

**Figure 6.** Estimated subjective annual occurrence probability between 1927 and 2016. The horizontal dashed line corresponds to the historical average value. Black bars represent the significant evolutions and dots indicate the occurrence of the following big earthquakes in California: 1933 Long Beach, 1952 Kern County, 1966 Parkfield, 1971 San Fernando, 1979 Coyotte Lake, 1979 Imperial Valley and Coyote Lake, 1983 Coalinga, 1984 Morgan Hill, 1987 Whittier Narrows, 1989 Loma Prieta, 1992 Big Bear and Landers, 1994 Northridge, 2003 San Simeon, 2004 Parkfield and 2014 Napa.

warnings of some earthquake researchers like Charles Richter (Geschwind 2001).

Regarding the second one, most of homeowners were only partially refunded, due to large deductibles, and decided to cancel their earthquake insurance policy (Meltsner 1978; Palm and Hodgson 1992b; Burnett and Burnett 2009). Moreover, the 1989 Loma Prieta earthquake struck just after a high seismic activity period during which $r_N$ has increased significantly. Indeed, the number of earthquakes occurring during the preceding 5 years with a moment magnitude greater than M5 reached the highest peak since 1855 between 1984 and 1998, as illustrated in Figure 7. The sequence includes in particular the 1983 Coalinga and the 1987 Whittier Narrows earthquakes. This specific temporal burst in seismic activity may have participated in homeowners' rising risk awareness (Table 4).

Some major socio-economic events also had an impact on $r_N$ (Table 4), at the light of the 1929 Great Depression and the Post World War II economic expansion. Insurance regulation acts also have an impact. During the period 1984-1985, the California Senate Assembly voted the Assembly Bill AB2865 (McAlister 1984) which required insurance companies to offer

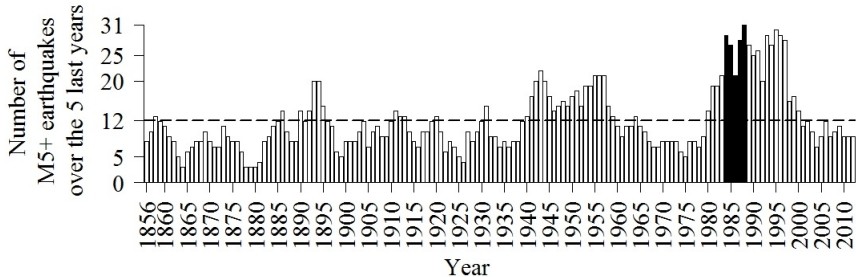

**Figure 7.** Number of earthquakes with a moment magnitude M5+ occurring during the 5 previous years. For year $N$, the sum is calculated from $N-5$ to $N-1$. Earthquakes occurring the same day count as one. Black bars correspond to the peak of seismicity observed between 1984 and 1988. The dotted line represents the average value from 1855 to 2012. The historical earthquake database used is taken from the UCERF3 (Field et al. 2013).

earthquake coverage. A second bill in 1995, named AB1366 (Knowles 1995), amended the Insurance Code Section 10089 to restrict the extent of the mandatory cover and addressed the insurance crisis caused by the 1994 Northridge earthquake. The new "mini-policy", met a limited success mainly because the price was more expensive for a lower guarantee (Reich 1996). Furthermore, the creation of the CEA was at the opposite of the trend in insurance market privatization and people assimilated it to an insurance industry bailout (Reich 1996). The combined impact on $r_N$ of the two Assembly Bills is null as $(1+30\%) \times (1-23\%) = 1$ (Table 4). It means that the efforts to promote earthquake insurance among the population by a first Assembly Bill was cancelled 20 years later by the other. Last, the discredit of a mistaken prediction can lead to a decrease of $r_N$, as it happened in 1928 about the occurrence of a major earthquake in Southern California Geschwind 1997; Yeats 2001).

$r_N$ is then compared to a scientific-based earthquake annual occurrence probability to assess the level of the homeowners' risk perception. This requires to assess the annual probability for a homeowner to be affected by a destructive earthquake. For that, the Shakemap footprints, released by the U.S. Geological Survey (USGS), are used. A Shakemap footprint gives, for a historical earthquake, the modelled ground motions for several metrics, including the macroseismic intensity on the Modified Mercalli Intensity scale (MMI). In California, a total of 564 Shakemap footprints are available on the USGS website for the period 1952-2018 and the magnitude range M3-M7.3. According to the MMI scale, heavy damage can be observed for an intensity above or equal to VIII. Among the 564 historical earthquakes, the Shakemap footprints report that only 21 have caused such a high intensity and are labelled as destructive earthquakes. For each of the 21 Shakemap footprints, the area corresponding to an intensity above or equal to VIII was reported and the population living inside was estimated using the Global Settlement Population Grid from the European Commission (Table 5). Finally, for each year since 1952, the annual rate of people was calculated by dividing the number of people having experienced an intensity above or equal to VIII by the total population of California according to the Global Human Settlement Population Grid (European Commission). The arithmetic mean of the annual rates during the period 1952-2018 is equal to $r^{Hist} = 0.038\%$. In this study, we consider $r^{Hist}$ as the true average probability for a California homeowner to be affected by a destructive earthquake. Therefore, the risk has always been underestimated by California homeowners because $AWR_N$ was lower than $r^{Hist}$ since 1926 (Figure 6). The severity

**Table 5.** Earthquakes occurred in California since 1950 with a maximum macroseismic intensity on the Modified Mercalli Scale (MMI) modelled by the Shakemap program (U.S. Geological Survey) above or equal to VIII. The columns Area and Number of people affected give the size and the estimated population of the area affected above or equal to VIII, respectively. The population has been assessed using the Global Human Settlement Population Grid (European Commission).

| Epicenter area | Year | Magnitude | Area (km$^2$) | Number of people affected |
|---|---|---|---|---|
| Parkfield | 1966 | 6.1 | 22 | 4 |
| Borrego Mountain | 1968 | 6.6 | 352 | 1 |
| San Fernando | 1971 | 6.7 | 147 | 55,991 |
| Imperial Valley | 1979 | 6.5 | 385 | 19,280 |
| Coyote Lake | 1979 | 5.8 | 21 | 17 |
| Eureka | 1980 | 7.3 | 4 | 20 |
| Coalinga | 1983 | 6.3 | 151 | 2,043 |
| Chalfant Valley | 1986 | 6.2 | 70 | 278 |
| Elmore Ranch | 1987 | 6.0 | 97 | 7 |
| Superstition Hills | 1987 | 6.5 | 199 | 1 |
| Loma Prieta | 1989 | 6.9 | 458 | 15,801 |
| Petrolia | 1992 | 7.2 | 46 | 120 |
| Joshua Tree | 1992 | 6.2 | 57 | 1 |
| Big Bear | 1992 | 6.5 | 47 | 9,756 |
| Landers | 1992 | 7.3 | 1,093 | 20,863 |
| Northridge | 1994 | 6.6 | 537 | 630,602 |
| Hector Mine | 1999 | 7.1 | 271 | 24 |
| San Simeon | 2003 | 6.6 | 5 | 10 |
| Parkfield | 2004 | 6.0 | 5 | <1 |
| El Mayor – Cucapah | 2010 | 7.2 | 28 | 46 |
| South Napa | 2014 | 6.0 | 30 | 412 |

underestimation is quantified through the earthquake risk awareness ratio ($AWR_N$) defined as:

$$AWR_N = \frac{r_N}{r^{Hist}} = 2632 \times r_N \tag{20}$$

$AWR_N$ being estimated since 1926, the take-up rate model is generalized in the next section and used to understand the current low take-up rate.

## 5 Understanding the current low take-up rate

Introducing the earthquake risk awareness ratio ($AWR_N$), we redefine the take-up rate model (Equation 15) as follows:

$$t_N^{Estimated} = \min\left[\left(\frac{AWR_N}{2632 - AWR_N} \times \frac{K}{P_N}\right)^{1.07}; 100\%\right] \tag{21}$$

5    From the estimated values of $AWR_N$ (Equations 19 and 20) and $P_N$ (Table 2), Figure 8 shows the fit between $t_N^{Estimated}$ (Equation 21) and $t_N$ (Table 2). The goodness-of-fit is $\left(\mathrm{R}^2 = 0.99\right)$ and the modelling errors are below $\pm 3\%$, as represented

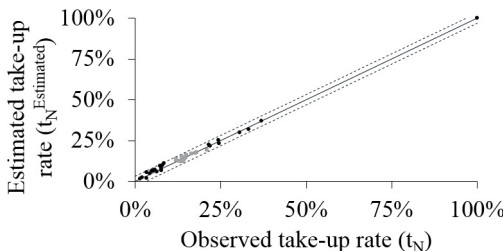

**Figure 8.** Comparison between the take-up rates observed and estimated using the model developed in this study. Grey points correspond to the period 1997-2016 which was used to calibrate the model. Dashed lines represent a buffer of $\pm 0.03$.

by the dashed lines. The record at 100% corresponds to the 1992 California Residential Earthquake Recovery program (CRER), which was a mandatory public earthquake insurance, including a cover amount for households up to $15,000 (USD 1992), at a very low premium amount (Lagorio et al. 1992).

Figure 9 illustrates the sensitivity of $t_N^{Estimated}$ to the average premium $P_N$ and the earthquake risk awareness ratio $AWR_N$ (Equation 20). The lines $AWR_N = 70\%$, $AWR_N = 100\%$ and $AWR_N = 425\%$ stand for the current situation, the true risk

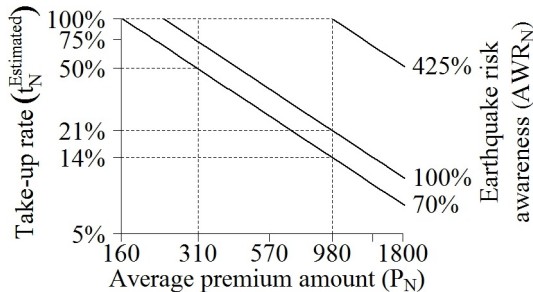

**Figure 9.** Relationship between the average premium amount, the take-up rate and the earthquake risk awareness. Financial values are in USD (2015).

5    level and the target value of $AWR_N$ for $t_N^{Estimated} = 100\%$, at the current price at $980 (USD 2015), respectively. According to the first one $(AWR_N = 70\%)$, half of homeowners $\left(t_N^{Estimated} = 50\%\right)$ are not willing to pay an average premium amount exceeding $P_N = \$310$ (USD 2015) per year, for an insurance cover. This would represent a 68% decrease in the price of earthquake insurance coverage.

On the opposite, the relatively low earthquake risk awareness $(AWR_N = 70\%)$, contributes only marginally to the current
10   low take-up rate $\left(t_N^{Estimated} = 14\%\right)$. In fact, even with the true level of risk $(AWR_N = 100\%)$, only $t_N^{Estimated} = 21\%$ of homeowners are expected to subscribe an insurance. This result is consistent with previous findings by Shenhar et al. (2015) who observed in Israel that the earthquake insurance take-up rate did not significantly increase after a large prevention

campaign.

To reach a 100% take-up rate with the current premium amount, according to the model, the earthquake risk awareness ratio has to reach 425%, which is very unlikely. Indeed, the UCERF3 assesses the annual probability of occurrence, in California, of an earthquake with a magnitude above M6.7 at 15% (Field et al. 2013). Under the assumption that the probability for a homeowner to experience a damaging earthquake is proportional to the occurrence of such an earthquake, reaching that awareness ratio would mean that homeowners consider this probability to reach the level of $425\% \times 15\% = 64\%$. In other words, only the belief that a destructive earthquake is imminent (i.e. a probability of occurrence somewhere in California during the year, for a 1994 Northridge-like earthquake, above 64%) can bring all homeowners to subscribe an earthquake insurance at the current price.

## 6   Conclusions

The model developed in the present study shows that the low earthquake insurance take-up rate, observed until 2016 for the Homeowners line of business in California, is due foremost to high premiums. Indeed, it assesses that no homeowners would prefer to stay uninsured against earthquake risk if the annual average premium would decrease from \$980 (as observed in 2016) to \$160 (USD 2015) or lower. Moreover, Kovacevic and Pflug (2011) have shown that a minimum capital is required to benefit from an earthquake insurance cover, otherwise the cost of the premium would drive to the ruin faster than holding the risk. They have also found that the lower the insurance premium the lower this minimum capital. Consequently, the results of Kovacevic and Pflug (2011) suggest that some uninsured people can be well aware of the earthquake risk but just cannot afford an earthquake insurance at the current price. This corroborates the result of the present study, because bringing such people to buy an earthquake insurance is foremost a matter of price, not of insufficient risk perception.

Nevertheless, as the current average premium amount corresponds to the annualized risk assessment, insurance companies would not have enough reserves to pay future claims following earthquakes if they collect on average only \$160 per policy (CEA 2017). Hence, the current insurance mechanism cannot meet the homeowners' demand and being sustainable, as it was the case for the 1992 CRER program (Kunreuther et al. 1998).

After this initiative, earthquake insurance has never been again mandatory in California. Because compulsory insurance is not only hard to implement, but also has an unpredictable impact (Chen and Chen 2013), new insurance solutions have emerged to increase the earthquake insurance market, like the parametric insurance. This insurance product stands out from traditional insurance policies by paying a fixed amount when the underlying metrics (e.g. the magnitude and the epicenter location) exceed a threshold, whatever the loss incurred by the policyholder. This new insurance claim process reduces significantly the loss uncertainty, the operating expenses and so the premium rate. In California, parametric insurance is offered to cover homeowners against earthquake risk since 2016 (Jergler 2017).

Nevertheless, risk awareness remains important in earthquake insurance consumption, and insurance companies also encourage local policies led by public authorities to improve prevention (Thevenin et al. 2018).

This study supports these initiatives by demonstrating that a new earthquake insurance scheme is necessary to meet the premium

expected by homeowners. However, it shows also that the lower the premium amount, the more the earthquake risk awareness contributes in insurance consumption (Fig. 9). Finally, only a global effort involving all stakeholders may fill the adoption gap in earthquake insurance coverage. This includes also the financial stakeholders, involved in the mortgage loans business. A recent study (Laux et al. 2016) have put in light that most of banks do not require an earthquake insurance from their mortgage loan clients, preferring instead to hold the risk and to increase the interest rate by +0.2% (in 2013). These additional bank fees are, on average, as expensive as the premium rate for the Residential earthquake insurance (0.185% in 2013 according to the CDI). Nevertheless, such a market practice puts an estimated $50bn-$100bn loss risk on the U.S. mortgage system (Fuller and Kang 2018). Consequently, expanding earthquake insurance cover, for homeowners, is also a challenge for the banking industry. In this respect, requiring an earthquake insurance for mortgage loans would strongly enhance the insurances' and public initiatives.

*Data availability.* Insurance market raw data and socio-economic indicators collected between 1921 and 2016 in California are available in the electronic supplement.

## Appendix A: The Expected Utility theory

The Expected Utility theory (also called the Von Neumann-Morgenstern Utilities), was introduced in 1947 by Von Neumann and Morgenstern. It models a decision-maker's preferences from a *basket of goods* (materialized by the function $g$), according to his *utility*. Under this framework, the *utility* is defined as the satisfaction that a decision-maker retrieves from the goods he gets. The *utility* is measured by an *utility function*, $U_i$, where $i$ is the name of the decision-maker. The function $U_i$ is defined by the Von Neumann-Morgenstern (1944) theorem which is based on several axioms that could be split in six parts (Narahari 2012). For the two firsts, let us considering a basket of three goods: $b = \{g_1; g_2 : g_3\}$. Then, the two first axioms state that:

1 **Completude**: the decision-maker can always compare two goods (e.g. $g_1$ and $g_2$) and decide which one he prefers or if the both are equal.

2 **Transitivity**: If the decision-maker prefers $g_1$ to $g_2$ and $g_2$ to $g_3$, then he also prefers $g_1$ to $g_3$.

Let us now considering that the decision-maker does not select a good from a basket of goods $b = \{g_1; g_3\}$ but instead participates to a lottery $l_1$ on $b$. We introduce $p_1$ and $1 - p_1$ as the probability for the decision-maker to win the good $g_1$ and $g_3$, respectively. Similarly, $l_2$ is the lottery on $b$ with the probabilities $p_2$ and $1 - p_2$. Here, the decision-maker can choose between two different lotteries. On this basis, the four remaining axioms are:

3 **Substitutability**: The decision-maker has no preference between $l_1$ and the same lottery where $g_1$ is changed by $g_4$ if the decision-maker has no preference between $g_1$ and $g_4$.

4 **Decomposability**: The decision-maker has no preference on the way to carry on the lottery $l_1$, as long as long as the probabilities ($p_1$ and $1 - p_1$) remain the same.

5 **Monotonicity**: Between $l_1$ and $l_2$, the decision-maker always prefer the one with the higher chance to win its favorite good (i.e. if he prefers $g_1$ to $g_3$, then the decision-maker prefers $l_1$ to $l_2$ when $p_1 > p_2$, and reciprocally).

6 **Continuity**: If the decision-maker prefers $g_1$ to $g_2$ and $g_2$ to $g_3$, it exists a value of $p_1$ for which the decision-maker has no preference between having $g_2$ and participating at $l_1$.

According to the Von Neumann-Morgenstern theorem, an utility function associates to a good $g_1$ a level of utility $U_i(g_1)$ for the decision-maker $i$. When the good $g_1$ is replaced by a lottery $l_1$, this theorem states that the level of utility is the weighted average: $U_i(l_1) = p_1 \times U_i(g_1) + (1 - p_1) \times U_i(g_3)$.

The first property of $U_i$ is to be strictly increasing (i.e. its first derivative $U_i'$ is strictly positive). When $g_1$ corresponds to the decision-maker's wealth, like in this study, it means that the decision-maker always wants to increase his level of wealth. At the opposite, the sign of the second derivative, $U_i''$, can be positive (i.e. $U_i$ is convex) or negative (i.e. $U_i$ is concave). It represents the decision-maker's behaviour: when $U_i'' < 0$, the decision-maker is risk-averse, meaning that the difference of utility between $g_1$ and $g_1 + 1$ is decreasing when $g_1$ increases. Still considering that $g_1$ is the decision-maker's wealth, it means that increasing

the wealth by $+€100$ provides a higher utility for a decision-maker with a wealth $g_1 = €1,000$ than for one with a wealth $g_1 = €1,000,000$. In terms of insurance, a risk-averse decsision maker want to be covered. Indeed, assuming that the decision-maker can be protected against losing $g_1$ with a probability $p_1$ when paying a premium amount $P_N = p_1 \times g_1$. Since the utility function is concave, the following equation is verified:

$$U_i(0) \times p_1 + U_i(g_1) \times (1 - p_1) < U_i(g_1 - P_N) \tag{A1}$$

Conversely, when $U_i'' > 0$, the decision-maker is risk-loving and behave in the opposite way.

Depending on the choice of $U_i$, this behaviour (risk-averse or risk-loving) can change depending on $g_1$. To represent it, the Arrow-Pratt measure of relative risk aversion (named after Pratt 1964 and Arrow 1965) is used:

$$RRA(g_1) = g_1 \times \frac{-U_i''(g_1)}{U_i'(g_1)} \tag{A2}$$

when RRA is decreasing, increasing or constant, it means that the decision-maker is less, more, or indifferently risk averse

with the value of the good $g_1$.

The utility function $U_i$ used in this study has been developed by Holt and Laury (2002).

$$U_i(g_1) = \begin{cases} \dfrac{g^{1-\beta_i}}{1 - \beta_i} & \text{if} \quad \beta_i \neq 1 \\ \log(g_1) & \text{else} \end{cases} \tag{A3}$$

Where $\beta_i$ is the *level of relative risk aversion* and materializes the decision-maker's risk behaviour from highly risk loving to highly risk averse. Indeed, one can verify that the *RRA* measure for the Holt and Laury (2002) utility function is: $RRA(g_1) =$

$\beta_i$. Therefore, this function belongs to the Constant Risk Relative Aversion utility function family. It means that for a decision-maker's behaviour is supposed uncorrelated to his wealth level ($g_1$). Furthermore, the parameter $\beta_i$ controls the decision-maker's behaviour (Table A1).

**Table A1.** Risk preference scale depending on the parameter $\beta_i$, according to Holt and Laury (2002).

| Risk preference classification | Range of relative risk aversion |
|---|---|
| highly risk loving | $\beta_i < -0.95$ |
| very risk loving | $-0.95 < \beta_i < -0.49$ |
| risk loving | $-0.49 < \beta_i < -0.15$ |
| risk neutral | $-0.15 < \beta_i < 0.15$ |
| slightly risk averse | $0.15 < \beta_i < 0.41$ |
| risk averse | $0.41 < \beta_i < 0.68$ |
| very risk averse | $0.68 < \beta_i < 0.97$ |
| highly risk averse | $0.97 < \beta_i < 1.37$ |
| stay in bed | $1.37 < \beta_i$ |

## Appendix B: Solution of the expected utility maximization equation

Let us consider the following mathematical problem in (Equation 14):

$$t_N^{Estimated} = \underset{0 \le t_N \le 1}{\operatorname{argmax}} \ \mathbb{E}\left[U(g_N(t_N))\right] \tag{14}$$

The function $f(t_N) = \mathbb{E}\left[U(g_N(t_N))\right]$ can be rewritten by detailing the expression of the expected utility $\mathbb{E}$:

$$f(t_N) = U(g_N(t_N)) \times \mathbb{P}\left(\mathcal{B}_{EQ}(r_N) = 0\right) + U(g_N(t_N)) \times \mathbb{P}\left(\mathcal{B}_{EQ}(r_N) = 1\right) \tag{14a}$$

Introducing the definition of $g_N(t_N)$ (Equation 13) gives:

$$f(t_N) = U(K - t_N \times P_N) \times \mathbb{P}\left(\mathcal{B}_{EQ}(r_N) = 0\right) + U(K - t_N \times P_N - K \times (1 - t_N)) \times \mathbb{P}\left(\mathcal{B}_{EQ}(r_N) = 1\right) \tag{14b}$$

5   Using next the definition of the function $U$ (Equation 12), $f(t_N)$ becomes:

$$
\begin{aligned}
f(t_N) &= \left(\frac{(K - t_N \times P_N)^{1-\beta}}{1-\beta}\right) \mathbb{P}\left(\mathcal{B}_{EQ}(r_N) = 0\right) + \left(\frac{(t_N \times (K - P_N))^{1-\beta}}{1-\beta}\right) \mathbb{P}\left(\mathcal{B}_{EQ}(r_N) = 1\right) \\
&= \frac{1}{1-\beta}\left[(1-r_N)(K - t_N \times P_N)^{1-\beta} + r_N(K - P_N)^{1-\beta} t_N^{1-\beta}\right]
\end{aligned}
\tag{14c}
$$

Furthermore, the derivative of $f(t_N)$ is equal to:

$$
\begin{cases}
\dfrac{\delta f(t)}{\delta t_N} = (-P_N)(1-r_N)(K - t_N \times P_N)^{-\beta} + r_N(K - P_N)^{1-\beta} t_N^{-\beta} \\
\dfrac{\delta f(0)}{\delta t_N} = +\infty \\
\dfrac{\delta f(1)}{\delta t_N} = (K - P_N)^{-\beta}(K r_N - P_N)
\end{cases}
\tag{14d}
$$

Thus, when $\frac{\delta f(1)}{\delta t_N} < 0$; $t_N^{Estimated}$ is equal to (otherwise, $t_N^{Estimated} = 1$):

$$
\frac{\delta f\left(t_N^{Estimated}\right)}{\delta t} = 0 \quad \Leftrightarrow \quad \left(\frac{K}{t_N^{Estimated}} - P_N\right)^{-\beta} \times P_N \times (1 - r_N) = r_N \times (K - P_N)^{1-\beta}
$$
$$
\Leftrightarrow t_N^{Estimated} = \frac{KX}{K - P_N + P_N X} \quad \text{where} \quad X = \left(\frac{r_N(K - P_N)}{P_N(1 - r_N)}\right)^{\frac{1}{\beta}}
\tag{14e}
$$

As the expression of $t_N^{Estimated}$ is complex, it can be simplified using the approximation: $\frac{P_N}{K} \approx 0$:

$$
\begin{aligned}
t_N^{Estimated} & = X \\
& = \left( \frac{r_N(K - P_N)}{P_N(1 - r_N)} \right)^{\frac{1}{\beta}} \\
& = \left( \frac{r_N}{(1 - r_N)} \times \frac{K}{P_N} \right)^{\frac{1}{\beta}} \times \left( 1 - \frac{P_N}{K} \right)^{\frac{1}{\beta}} \\
& = \left( \frac{r_N}{(1 - r_N)} \times \frac{K}{P_N} \right)^{\frac{1}{\beta}}
\end{aligned}
\tag{14f}
$$

*Author contributions.* Adrien Pothon had the following role in this study: Conceptualization, Formal analysis, Investigation, Methodology and Writing - original draft. Philippe Gueguen performed the Supervision, Validation, Visualization roles and also wrote the original draft. Pierre-Yves Bard and Sylvain Buisine were involved in the Validation as well as in the writing - review & editing process.

*Competing interests.* The authors declare that they have no conflict of interest.

*Acknowledgements.* The authors acknowledge Madeleine-Sophie Déroche, CAT Modelling R&D Leader at AXA Group Risk Management, for her contribution to this work. They are also grateful to the California Department of Insurance, the EM-DAT, Robert Shiller, the U.S. Census, the U.S. Department of Labor, and the USGS for providing an open-access to their databases. Philippe Guéguen has received funding from the European Union's H2020 research and innovation programme under the Marie Sklodowska-Curie grant agreement N° 813137 and was supported by funding from Labex OSUG@2020 (Investissements d'avenir, ANR10-LABX56).

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
