# Peer review of "California earthquake insurance unpopularity: the issue is the price, not the risk perception."

_Natural Hazards and Earth System Sciences, 2019_

## Referee Comment (RC1) · Anonymous Referee #1 · 20 Feb 2019

1. I enjoyed the paper. 2. The conclusions are realistic 3. The math is a bit obtuse and a few more words of explanation may help the non-mathematics types 4. It is an intractable problem - and the reasons for the poor uptake matches my previous experience

---

## Referee Comment (RC2) · Anonymous Referee #2 · 12 Jun 2019

The authors present a very concise and timely study on the willingness to underwrite earthquake risk insurance, taking California as an example. In general, the article is well-written, the methods are very clear and the results provide insightful information on risk perception and economic behaviour. The overall topic is within the scope of the target journal.

I only have three small comments:

The authors state in their introduction that "...even if seismic risk is not taken into account into real estate value assessment (Porter et al. 2004), a difference between the value of a property and its replacement cost after an earthquake can bring negotiations on the claim amounts, and therefore have an impact on the insurance demand (Garratt and Marshall 2003)." It would be interesting to add here some explanatory sentences

since this seems to be exceptional in comparison to other natural hazard insurances, such as e.g. discussed by Holub and Fuchs (2009) for European mountain regions or some other works with respect to the situation in Switzerland (Röthlisberger et al.'s works) France and England. In the latter studies, it is the philosophy of the insurer to provide reconstruction values and not market prices as a basis for calculation of premiums in order to avoid taking into account too high market value fluctuations in some areas. This can also be valid for some of the expensive villages in California, and may have an influence on the results.

Second, information such as "data volume before the 90s" should be changed to "...1990s" to clearly state the reference (year, decade).

Third, the paper would heavily benefit from a bit more explanation on the equations used – information could be presented in an Appendix.

Please note, that the indicated references is for illustration purpose only, this does not mean that authors should include the item in a revised version.

References mentioned:

Holub, M., and Fuchs, S.: Mitigating mountain hazards in Austria – Legislation, risk transfer, and awareness building, Natural Hazards and Earth System Sciences, 9, 523-537, https://doi.org/10.5194/nhess-9-523-2009, 2009.

Röthlisberger, V., Zischg, A., and Keiler, M.: Identifying spatial clusters of flood exposure to support decision making in risk management, Science of the Total Environment, 598, 593-603, 2017.

Röthlisberger, V., Zischg, A., and Keiler, M.: A comparison of building value models for flood risk analysis, Natural Hazards and Earth System Sciences, 18, 2431-2453, https://doi.org/10.5194/nhess-18-2431-2018, 2018.

2019-29, 2019.

---

## Author Comment (AC2) · 23 Jul 2019

Dear Anonymous Referee,

Thank you for your nice comment on our paper. All your suggestions have been taken into account for improving the clarity of the paper. Furthermore, additional references that you mention are very interesting and show that insurance faces several issues not only for earthquake risk but also other natural catastrophes like flood.

Best regards,

Adrien Pothon
* * *
[Figure]

2019-29, 2019.

---

## Author Comment (AC3) · 23 Jul 2019

The comment was uploaded in the form of a supplement:
https://www.nat-hazards-earth-syst-sci-discuss.net/nhess-2019-29/nhess-2019-29-AC3-supplement.pdf
* * *

---

## Author Response (AR1)

**California earthquake insurance unpopularity: the issue is the price, not the risk perception.**

Adrien Pothon[1,2], Philippe Gueguen[1], Sylvain Buisine[2], and Pierre-Yves Bard[1]

[1]ISTerre, Université de Grenoble-Alpes/Université de Savoie Mont Blanc/CNRS/IRD/IFSTTAR, Grenoble, France
[2]AXA Group Risk Management, Paris, France

**Correspondence:** Adrien Pothon (adrien.pothon@univ-grenoble-alpes.fr)

**POINT-BY-POINT RESPONSE TO THE REVIEWS**

**Anonymous Referee #1**

(R) **Reviewer:** The math is a bit obtuse and a few more words of explanation may help the non-mathematics types

(A) **Answer:** The paper has been improved with more explanations on math (Change #2 & #3)

5 **Anonymous Referee #2**

(R) **Reviewer:** The authors state in their introduction that ". . .even if seismic risk is not taken into account into real estate value assessment (Porter et al. 2004), a difference between the value of a property and its replacement cost after an earthquake can bring negotiations on the claim amounts, and therefore have an impact on the insurance demand (Garratt and Marshall 2003)." It would be interesting to add here some explanatory sentences since this seems to be exceptional
10 in comparison to other natural hazard insurances, such as e.g. discussed by Holub and Fuchs (2009) for European mountain regions or some other works with respect to the situation in Switzerland (Röthlisberger et al.'s works) France and England. In the latter studies, it is the philosophy of the insurer to provide reconstruction values and not market prices as a basis for calculation of premiums in order to avoid taking into account too high market value fluctuations in some areas. This can also be valid for some of the expensive villages in California, and may have an influence on the
15 results.

(A) **Answer:** Some explanatory sentences have been added to explain in what extent earthquake risk has not a major impact on real estate values (Change #1)

(R) **Reviewer:** Second, information such as "data volume before the 90s" should be changed to ". . .1990s" to clearly state the reference (year, decade).

20 (A) **Answer:** The modification proposed has been made.

(R) **Reviewer:** Third, the paper would heavily benefit from a bit more explanation on the equations used – information could be presented in an Appendix.

(A) **Answer:** More details about the Ruin Theory is provided in a new Appendix (Change #4)

**LIST OF ALL RELEVANT CHANGES MADE IN THE MANUSCRIPT**

**Change #1**

**Page 2, Line 1-4:** Furthermore, even if seismic risk is not taken into account into real estate value assessment (Porter et al. 2004), a difference between the value of a property and its replacement cost after an earthquake can bring negotiations on the claim amounts, and therefore have an impact on the insurance demand (Garratt and Marshall 2003).

**has been replaced by:**

**Page 2, Line 1-14:** Pricing methods used in the real estate market, do not take into account seismic risk. Indeed, based on data from the World Housing Encyclopedia, Pothon et al. (2018) have shown that building construction price is not correlated to the seismic vulnerability rating. Moreover, earthquake insurance is not mandatory in California for residential mortgage (CDI 2019, Information Guides, Earthquake Insurance). Still, lenders for commercial mortgages, are used to requiring an earthquake insurance only when the probable maximum loss is high (Porter et al. 2004). However, Porter et al. (2004) have shown that taking into account earthquake risk for calculating the building's net asset value can have a significant impact in earthquake prone areas like California.

Also, insured people against earthquake risk can receive a compensation lower than the loss incurred because most of earthquake insurance policies in California include a deductibles amount (i.e. at the charge of the policyholder) and are calibrated based on a total reconstruction cost, declared by the policyholder. As reported by Marquis et al. (2017) after the 2010–2011 Canterbury (NZL) earthquakes, this amount can be inadequate for the actual repairing costs. Both high deductibles amount (Meltsner 1978; Palm and Hodgson 1992b; Burnett and Burnett 2009) and underestimated total reconstruction cost (Garatt and Marshall 2003) can make insured people felt unprotected after an earthquake, disreputing this kind of insurance.

**Change #2**

**Page 6, Line 15-18:** where $\Delta_X^Y$ is equal to the ratio $Y$ over $X$. Furthermore, by definition of $W_N^{Res}$, $P_N^{Res}$ and $t_N^{Res}$, the combined variation is:

$$\Delta_{W_{N-1}^{Res}}^{W_N^{Res}} = \Delta_{Fi_{N-1}^{Res}}^{Fi_N^{Res}} \times \Delta_{t_{N-1}^{Res}}^{t_N^{Res}} \times \Delta_{P_{N-1}^{Res}}^{P_N^{Res}} \tag{1}$$

As we have no data regarding $\Delta_{Fi_{N-1}^{Res}}^{Fi_N^{Res}}$ before 1996 (Table 1), it is assumed as equal to the variation of the California Population $Pop_N$:

**has been replaced by:**

**Page 7, Line 1-4:** Furthermore, the variation of the total written premium for the Residential line of business $\left(\Delta^{W_N^{Res}}_{W_{N-1}^{Res}}\right)$ is linked to $\Delta^{P_N^{Res}}_{P_{N-1}^{Res}}$, $\Delta^{t_N^{Res}}_{t_{N-1}^{Res}}$ and $\Delta^{Fi_N^{Res}}_{Fi_{N-1}^{Res}}$ as follows:

$$
\begin{aligned}
\Delta^{W_N^{Res}}_{W_{N-1}^{Res}} &= \frac{W_N^{Res}}{W_{N-1}^{Res}} \\
&= \frac{P_N^{Res} \times Nb_N^{Res}}{P_{N-1}^{Res} \times Nb_{N-1}^{Res}} \\
&= \frac{P_N^{Res} \times Fi_N^{Res} \times t_N^{Res}}{P_{N-1}^{Res} \times Fi_{N-1}^{Res} \times t_{N-1}^{Res}} \\
&= \Delta^{P_N^{Res}}_{P_{N-1}^{Res}} \times \Delta^{Fi_N^{Res}}_{Fi_{N-1}^{Res}} \times \Delta^{t_N^{Res}}_{t_{N-1}^{Res}}
\end{aligned}
\tag{2}
$$

As we have no data regarding the annual variation of the number of fire insurance policy for the residential line of business

5   $\left(\Delta^{Fi_N^{Res}}_{Fi_{N-1}^{Res}}\right)$ before 1996 (Table 1), it is assumed as equal to the variation of the California Population $Pop_N$:

**Change #3**

**Page 10, Line 17:**

$$
\frac{W_N/Pop_N}{W_{N-1}/Pop_{N-1}} = \left(\frac{P_N}{P_{N-1}}\right)^{-0.07} \times \left(\frac{r_N \times (1-r_{N-1})}{r_{N-1} \times (1-r_N)}\right)^{1.07}
\tag{3}
$$

**has been replaced by:**

**Page 11, Line 6-9:**

$$

[revised manuscript text omitted]